# PDZK1-Interacting Protein 1(PDZKIP1) Inhibits Goat Subcutaneous Preadipocyte Differentiation through Promoting Autophagy

**DOI:** 10.3390/ani13061046

**Published:** 2023-03-14

**Authors:** Dingshuang Chen, Yanyan Li, Tingting Hu, Chengsi Gong, Guangyu Lu, Xiaotong Ma, Yong Wang, Youli Wang, Yaqiu Lin

**Affiliations:** 1College of Animal Science and Veterinary, Southwest Minzu University, Chengdu 610041, China; 2Key Laboratory of Qinghai-Tibetan Plateau Animal Genetic Resource Reservation and Utilization, Ministry of Education, Southwest Minzu University, Chengdu 610041, China; 3Key Laboratory of Sichuan Province for Qinghai-Tibetan Plateau Animal Genetic Resource Reservation and Exploitation, Southwest Minzu University, Chengdu 610041, China

**Keywords:** PDZK1IP1, adipogenesis, autophagy, goat, preadipocytes

## Abstract

**Simple Summary:**

PDZK1-interacting protein 1 (PDZK1IP1) is a membrane-associated non-glycosylated protein and is involved in development and tumorigenesis. However, the role of PDZK1IP1 in goat subcutaneous preadipocyte differentiation is unknown. In this work, we found that PDZK1IP1 acts as a regulator of adipogenesis, and inhibits goat subcutaneous preadipocyte differentiation by promoting autophagy. The results will help to better understand the biological functions of PDZK1IP1 in goat subcutaneous preadipocytes and improve studies of its molecular mechanism.

**Abstract:**

PDZK1IP1 is highly expressed in tumor tissue and has been identified as a tumor biomarker. However, the role of PDZK1IP1 in goat subcutaneous preadipocyte differentiation remains largely unknown. The molecular mechanism of autophagy in regulating the differentiation of goat subcutaneous preadipocytes has not been clarified yet. In our study, PDZK1IP1 gain of function and loss of function were performed to reveal its functions in preadipocyte differentiation and autophagy. Our results showed that the overexpression of PDZK1IP1 inhibited the differentiation of goat subcutaneous preadipocytes, whereas it promoted autophagy. Consistently, the knockdown of PDZK1IP1 demonstrated the opposite tendency. Next, we investigated whether PDZK1IP1 inhibited the differentiation of goat preadipocytes by regulating autophagy. We found that inhibiting autophagy can rescue the PDZK1IP1-induced differentiation restraint in goat subcutaneous preadipocytes. In conclusion, PDZK1IP1 acts as a regulator of adipogenesis, and inhibits goat subcutaneous preadipocyte differentiation through promoting autophagy. Our results will contribute to further understanding the role and mechanism of PDZK1IP1 in controlling adipogenesis.

## 1. Introduction

Adipose tissue is of high significance for numerous mammals and plays a critical role in regulating lipid homeostasis and the energy balance in the body [1,2,3]. Adipose tissue in mammals can be classified into visceral fat (VAT), subcutaneous fat (SAT) and intramuscular fat (IMF) according to its location. In different sites of the body, these adipose tissue types have different biological functions. Subcutaneous fat (SAT) and intramuscular fat (IMF) in humans are associated with multiple metabolic dysfunctions, such as obesity and diabetes [4]. Visceral fat (VAT), compared with subcutaneous fat (SAT) and intramuscular fat (IMF), is viewed to be the more adverse adipose tissue depot and has been linked to dyslipidemia, cardiovascular diseases and insulin resistance [5,6,7]. However, subcutaneous fat (SAT) and intramuscular fat (IMF) in farm animals have drawn increased attention, because they are an important factor that impacts meat quality [8]; for example, high intramuscular fat content could increase meat tenderness, flavor and juiciness to improve meat quality. However, increased subcutaneous fat content can reduce the lean meat ratio of the carcass, reducing the economic value of the meat [9,10]. Therefore, shedding light on the lipid metabolism mechanism is crucial for human diseases and agricultural economy.

Autophagy is an essential self-protection mechanism that maintains cellular homeostasis by eliminating excess or damaged organelles, proteins and macromolecules [11]. It has been shown that autophagy is involved in the differentiation of several cell types, including preadipocytes. In 3T3-L1 preadipocytes, knockdown of autophagy-related gene 7 (Atg7), which is a key autophagy gene, will decrease the protein levels of preadipocyte differentiation markers and inhibit lipid accumulation. An adipocyte-specific ATG7-knockout mice model displayed a lean phenotype and decreased white adipose tissue (WAT) mass [12,13]. In addition, autophagy is upregulated in the adipose tissue of human obese patients [14,15]. The above studies provide strong evidence that autophagy plays a vital role in preadipocyte differentiation.

PDZK1-interacting protein 1 (PDZK1IP1), a 17 kDa insoluble and non-glycosylated protein, is also termed MAP17, SPAP and DD96 [16,17]. It was initially identified as an epithelium-specific molecule and later found to be highly expressed in breast, cervical, ovarian and prostate tumors [18,19]. Moreover, there is evidence that PDZK1IP1 has a widespread impact on tumor cell biological functions, including proliferation, apoptosis, migration, invasion and so on [20,21,22]. This shows that PDZK1IP1 is highly associated with cancer progression. An increasing number of reports regarding the functions and influences of PDZK1IP1 on multiple cancers have been released, and in depth mechanistic studies are also ongoing. However, the regulating effects of PDZK1IP1 in regulating lipid deposition in preadipocytes have not been reported. In our previous study, we successfully cloned the goat PDZK1P1 gene sequence and demonstrated that PDZK1IP1 overexpression could promote goat subcutaneous preadipocyte proliferation [23]. In addition, it is worth noting that bortezomib can induce autophagy in breast cancer cells, while overexpressed PDZK1IP1 can inhibit autophagy in these cells [24]. This finding indicates that PDZK1IP1 could regulate autophagy. As mentioned previously, autophagy is involved in the differentiation of preadipocytes. Therefore, we wish to determine whether PDZK1IP1 affects the autophagy and differentiation of subcutaneous preadipocytes of goats, and whether autophagy regulated by PDZK1IP1 is correlated with adipocyte differentiation.

Thus, the current study sought to investigate the underlying roles and potential mechanisms of PDZK1IP1 in goat subcutaneous preadipocyte lipid accumulation. Our study focused on the effects of PDZK1IP1 on goat subcutaneous preadipocyte differentiation via in vitro adipocyte models.

## 2. Materials and Methods

### 2.1. Animals and Tissue Collection

In this study, animal samples, which were from Jianzhou Daer goats (*Capra hicus*), were approved by Sichuan Jianzhou Dageda Aminal Husbandry Co., Ltd. (Jianyang, Sichuan, China). The body weight of goats was approximately 50 kg. In addition, these goats were given ad libitum access to the same diet and water before slaughter. A total of three Jianzhou Daer goats (*Capra hicus*), male, one year old, were slaughtered. Tissue samples including heart, liver, spleen, lungs, kidney, longissimus dorsi, biceps femoris, arm triceps, abdominal fat and subcutaneous fatty tissue were harvested and immediately frozen in liquid nitrogen. The slaughter procedure was conducted at the College of Animal Science and Veterinary, Southwest Minzu University. All experimental procedures were approved by the Institutional Animal Care and Use Committee of Southwest Minzu University (Chengdu, Sichuan, China) with approval No. 2020086, 2020.

### 2.2. Cell Isolation and Cell Culture

Goat subcutaneous preadipocytes were isolated and cultured, as previously reported [25]. Briefly, in our previous study, subcutaneous adipose tissue was collected under sterile conditions from the backs of 7-day-old goats (*n* = 3, male) and was cut with scissors into small pieces. The minced tissue was digested with collagenase type I at 37 °C for 1 h. After digestion, these samples were filtered (75-μm filter) and centrifuged (2000 r/min for 5 min). Following this, red blood cell lysis was performed to remove red blood cells, and the cell suspensions were centrifuged (1500 r/min for 5 min) again. Then, the cell precipitates were re-suspended in DMEM/F12 (Hyclone, Logan, UT, USA) containing 10% FBS (Gemini, Calabasas, CA, USA) and 1% biresistance (Gemini, Calabasas, CA, USA). The cell suspension was seeded on 60-mm dishes and cultured in the incubator (37 °C, 5% CO_2_).

### 2.3. Cell Transfection

Goat subcutaneous preadipocytes were seeded in 6-well or 24-well plates and transfected using TurboFect Transfection Reagent (Thermo, Waltham, MA, USA) according to the manufacturer’s instructions. For transfection in 6-well plates, the transfection mixture comprising 2 μg plasmid, 6 μL of transfection reagent and 400 μL of Opti-MEM (Gibco, Calabasas, CA, USA) was added to each well. Likewise, transfection mixture including 6 μL siRNA (20 µM), 6μL of transfection reagent and 400 μL of Opti-MEM (Gibco, Calabasas, CA, USA) was added to each well. The PDZK1IP1-overexpressing plasmid vector (PDZK1IP1) was previously constructed and saved in our lab. The Si-PDZK1IP1 directed against PDZK1IP1 was purchased from GnenPharma (GnenPharma, Shanghai, China). The sequences of Si-PDZK1IP1 and the Si negative control (si-NC) were as follows:Si-PDZK1IP1:forward strand: 5′-GAGAAUGCCUAUGAGAACATT−3′.reverse strand: 5′-UGUUCUCAUAGGCAUUCUCTT−3′.Si-NC:forward strand: 5′-CAAUCGCCUUUGCUGUCAATT−3′.reverse strand: 5′-ACGUGACACGUUCGGAGAATT−3′.

### 2.4. Induced Differentiation of Goat Subcutaneous Preadipocytes

Goat subcutaneous preadipocytes in the logarithmic growth phase were harvested for differentiation induction. Goat subcutaneous preadipocytes were seeded on 6-well plates at the density of 8 × 10^4^ cells/well. At 24 h after transfection, medium was replaced with adipocyte-induction medium (DMEM/F12 medium containing 10% FBS, 1% penicillin and streptomycin and 50 µmol/L oleic acid) and the cells were cultured for an additional 48 h. Then, the cells were harvested 48 h after the induction of differentiation and used in the subsequent experiments.

### 2.5. Oil Red O and Bodipy Staining

For Oil Red O staining and Bodipy staining, cells were seeded in 24-well plates at a density of 1 × 10^4^ cells per well. Then, 24 h after transfection, goat subcutaneous preadipocytes were induced in complete medium with oleic acid (50 µM) for up to 48 h. The 24-well plate was removed, the supernatant was discarded, and 1 mL pre-chilled PBS was added to wash cells in each well. At room temperature, goat subcutaneous adipocytes were fixed with 4% paraformaldehyde for 15 min and then washed with pre-chilled PBS twice. Next, cells were stained using Oil Red O or Bodipy working solution for 20 min, and then washed twice in pre-chilled PBS. Lastly, goat subcutaneous adipocytes were observed and photographed under an Olympus IX-73 microscope (Tokyo, Japan). Staining was quantified by adding 100% isopropanol into each of the 24 wells to dissolve Oil Red O and measuring the absorbance at 490 nm.

### 2.6. AO and MDC Staining

Detection of autophagy was performed by using an AO staining kit (Solarbio, Beijing, China) and MDC staining kit (Solarbio, Beijing, China) according to the manufacturer’s instructions. In brief, goat subcutaneous preadipocytes were transfected with plasmid or Si-RNA for 48 h and then were fixed with 4% paraformaldehyde for 15 min. After fixation, AO working solution (1 mg/mL) and MDC working solution were used at room temperature to stain the cells for 30 min. The staining then was observed under a fluorescence microscope and photographed (magnification: 400×).

### 2.7. Western Blotting

The total proteins of goat subcutaneous preadipocytes were extracted via RIPA cell lysis buffer on ice for 30 min, followed by centrifugation at 12,000 rpm for 10 min at 4 °C. The total protein in each sample was detected by using the BCA Protein Assay Kit (Biosharp, Shanghai, China), and denatured protein (20 µg/lane) from each sample was subjected to 12% SDS-PAGE. The electrophoresed proteins were subsequently transferred to a 0.22 µm PVDF membrane with 5% nonfat milk at room temperature for 2 h. Then, the membranes were incubated with anti-LC3 (1:500, Wanleibio, Jilin, China, bs−5713R), anti-p62 (1:500, Wanleibio, Jilin, China, bs−5713R) and anti-β-actin (1:5000, Abways, Shanghai, China, AB0035) at 4 °C overnight. After washing, the membranes were washed in TBST buffer and horseradish peroxidase-labeled secondary antibodies (1:5000, Abways, Shanghai, China, AB0102) were incubated for 2 h. Finally, we used ECL (Bio-Rad, Hercules, CA, USA) to detect target proteins.

### 2.8. Real-Time Quantitative PCR (qRT-PCR) Analysis

Total RNA was isolated from goat subcutaneous preadipocytes with an RNAiso Plus reagent (Takara, Dalian, China), according to the manufacturer’s instructions. Then, 1 μg total RNA was reverse-transcribed using the RevertAid First Strand cDNA Synthesis Kit (Thermo, Waltham, MA, USA) and real-time PCR (qRT-PCR) was subsequently performed using the SYBR Green Premix Ex Taq Kit (TaKaRa, Japan) on the BioRad Real-Time PCR System. All of the primer sequence information in the study is presented in Table 1 and all gene expression analyses were calculated using the (2^−△△ct^) method [26].

### 2.9. Statistical Analysis

All data were analyzed using the GraphPad Prism 9 software. For comparisons between two groups or multiple groups, Student’s two-tailed *t*-test or multiple comparison test was performed to evaluate the significant differences. In every figure, significance is represented with asterisks (* *p* < 0.05; ** *p* < 0.01). The number of biological repeats in all bars was three.

## 3. Results

### 3.1. The Expression Level of PDZKIP1 Was Downregulated during the Differentiation of Goat Subcutaneous Preadipocytes

To probe the role of PDZK1IP1 in goat subcutaneous preadipocytes, we first examined its expression in various goat tissue samples. The results of qRT-PCR analysis showed that PDZK1IP1 mRNA was expressed in all the tested tissue samples. The expression level of PDZK1IP1 was higher in the kidneys and lower in abdominal fat compared to the expression level in the heart. In addition, PDZK1IP1 was also expressed in subcutaneous adipose tissue (Figure 1A). Because subcutaneous fat deposition is crucial for meat quality, we wished to investigate whether PDZK1IP1 is involved in the regulation of subcutaneous fat deposition. Subsequently, we isolated goat subcutaneous preadipocytes and induced differentiation with 50 µmol/L oleic acid for 4 days. Goat subcutaneous preadipocyte differentiation was evaluated by Oil Red O staining, and the results showed that both the size and the number of lipid droplets were increased as the induction time was prolonged (Figure 1B,C). At the same time, we measured the expression of three differentiation marker genes, PPARγ, C/EBPα and C/EBPβ, by qRT-PCR. We found that the expression of these three markers was increased during preadipocyte differentiation induction (Figure 1D–F). The above morphological and molecular detection results indicated that goat subcutaneous preadipocytes were successfully isolated. Next, we examined the expression level of PDZK1IP1 during goat subcutaneous preadipocyte differentiation by qRT-PCR. The result demonstrated that the expression of PDZK1IP1 was downregulated in differentiated adipocytes (Figure 1G). These results indicated that PDZK1IP1 may be involved in the process of goat subcutaneous preadipocyte differentiation.

### 3.2. PDZK1IP1 Functioned as a Repressor of Goat Subcutaneous Preadipocyte Differentiation

From the above study, we sought to determine whether PDZKIP1 is a regulator of goat subcutaneous preadipocyte differentiation; the PDZK1IP1 expression plasmid (PDZK1IP1) or an empty vector (Vector) was transiently transfected into goat subcutaneous preadipocytes for 48 h. qRT-PCR examination showed that the cells transfected with PDZK1IP1 expression plasmid could significantly increase their PDZK1IP1 expression when compared to the cells transfected with the control vector (Figure 2A). Subsequently, as judged by Oil Red O staining, the semi-quantitative assessment of Oil Red O content and Bodipy staining, the overexpression of PDZK1IP1 can significantly decrease lipid droplet accumulation in goat subcutaneous adipocytes (Figure 2B–D). Additionally, the adipocyte differentiation markers, such as PPARγ, C/EBPα, C/EBPβ and SREBP1, were significantly downregulated in PDZK1IP1 overexpression cells (Figure 2E).

Nevertheless, we further confirmed the anti-adipogenic effect of PDZK1IP1 through loss of function studies, in which we hypothesized that PDZK1IP1 knockdown would enhance adipose conversion. We then transfected goat subcutaneous preadipocytes with specific Si-RNA to knock down PDZK1IP1. qRT-PCR was used to detect the expression level of PDZK1IP1 in the subcutaneous preadipocytes of goats. Compared with the control cells, its expression was significantly downregulated (Figure 3A). The loss of function assay found that Si-PDZK1IP1 cells displayed stronger adipogenesis potential, including more lipid droplets formed (Figure 3B–D), and the expression of adipocyte marker genes such as C/EBPα, C/EBPβ and AP2 was upregulated (Figure 3E). Taken together, these data suggested that PDZK1IP1 acts as a repressor to regulate goat subcutaneous preadipocyte differentiation.

### 3.3. PDZK1IP1 Positively Modulates Autophagy Activation in Goat Subcutaneous Preadipocytes

Studies have shown that the overexpression of PDZK1IP1 inhibits bortezomib-induced autophagy in breast cancer [24], while the effect of PDZK1IP1 on autophagy regulation in goat subcutaneous preadipocytes has not been reported before. When autophagy is activated, LC3I is converted into LC3II, which is localized to autophagosome membranes and is essential for autophagosome membrane biogenesis. Therefore, LC3II has been recognized as a biomarker of the formation of autophagosomes [27]. In addition to LC3II, p62 (a protein specifically degraded in lysosomes) is another widely used autophagy marker, which is downregulated in the autophagic process [28]. In order to clarify the role of PDZK1IP1 in goat subcutaneous preadipocyte autophagy, the expression levels of two autophagy-related proteins were analyzed firstly by Western blot assay. In our study, the Western blot experiment showed that PDZK1IP1 overexpression significantly increased the conversion of LC3I to LC3II and reduced the expression of p62, indicating that PDZK1IP1-induced autophagic and autophagic flux was unobstructed (Figure 4A, see Appendix A for all western original images). Then, acridine orange (AO) and monodansylcadaverine (MDC) were used to observe the autophagosome. We found that the number of fluorescent puncta was increased upon PDZK1IP1 overexpression (Figure 4B).

Next, a loss of function assay was performed with PDZK1IP1-specific siRNA in goat subcutaneous preadipocytes to investigate the effect of PDZK1IP1 on autophagy. After PDZK1IP1 was knocked down, LC3II accumulation and P62 expression were measured by Western blot to evaluate autophagy initiation and autophagy flux. Compared to the control cells, knockdown of PDZK1IP1 significantly decreased the LC3-II/I ratio and increased the p62 level, indicating that Si-PDZK1IP suppresses autophagy initiation and inhibits autophagic degradation (Figure 5A, see Appendix A for all western original images). Subsequently, AO and MDC staining also were performed to observe autophagosomes, which indicated that PDZK1IP1 knockdown decreased autophagosome formation compared with the control group (Figure 5B). This confirmed our conjecture. Altogether, our data suggested that PDZK1IP1 promotes autophagy formation in goat subcutaneous preadipocytes.

### 3.4. Inhibition of Autophagy Can Rescue PDZK1IP1-Induced Differentiation Restraint in Goat Subcutaneous Preadipocytes

Our study has shown that PDZK1IP1 inhibits the differentiation and promotes the autophagy of goat subcutaneous preadipocytes. Therefore, we considered whether PDZK1IP1 affects goat subcutaneous preadipocyte differentiation through the autophagy pathway and treated PDZK1IP1 overexpression cells and control cells with or without NH_4_CL during adipogenesis. We found that preventing the degradation of autophagosomes adequately increased the expression levels of LC3II and p62 (Figure 6A, see Appendix A for all western original images). Meanwhile, compared with the control cells without NH_4_CL, the number of autophagosomes was significantly increased (Figure 6B,C). These results also showed that NH_4_CL successfully inhibited autophagy and decreased the difference in autophagy between the PDZK1IP1 overexpression group and the control group. To further validate the role of PDZK1IP1-regulated autophagy in its inhibitry effects on adipogenic differentiation, next, we treated cells with 20 mM NH_4_CL for 48 h during goat subcutaneous preadipocyte differentiation. The efficiency of PDZK1IP1 overexpression was assessed by qRT-PCR, and results showed that PDZK1IP1 was successfully overexpressed in the cells treated with or without NH_4_CL (Figure 7A). In addition, we observed that NH_4_CL treatment could reverse the inhibition of lipid accumulation caused by PDZK1IP1 overexpression (Figure 7B–D). Consistent with the phenotype, the mRNA levels of adipocyte differentiation markers, including PPARγ, C/EBPα and C/EBPβ, but not SREBP1, were remarkably downregulated in PDZK1IP1 overexpression cells, which could be rescued to normal levels by NH_4_CL treatment (Figure 7E). In summary, these data demonstrate that PDZK1IP1 inhibits preadipocytes’ differentiation by promoting autophagy.

## 4. Discussion

In our previous studies, we have successfully cloned PDZK1IP1’s gene sequence and demonstrated that PDZK1IP1 could promote subcutaneous preadipocyte proliferation. However, we still lack knowledge of the effect of PDZK1IP1 on goat subcutaneous preadipocytes’ differentiation. Previously, it was shown that PDZK1IP1 was rarely expressed in normal tissue but highly expressed in tumor tissue, and PDZK1IP1 may be an oncogene that is correlated with cancer development [20,29,30]. Thus, we examined the expression of PDZK1IP1 in multiple goat tissue samples and found it to be expressed in all tested tissue types, including subcutaneous adipose tissue. We hypothesized that PDZK1IP1 may play a certain role in subcutaneous adipose tissue. We then examined the level of PDZK1IP1 expression in goat subcutaneous differentiated adipocytes and found that PDZK1IP1 expression was significantly changed. This forced us to determine whether PDZK1IP1 has an effect on goat subcutaneous preadipocyte differentiation and the specific mechanistic function of PDZK1IP1 in regulating goat subcutaneous preadipocyte differentiation. To finally confirm our conjecture, gain and loss of function experiments were performed in goat subcutaneous preadipocytes in vitro. We found that PDZK1IP1 could inhibit lipid droplet accumulation at the morphological level. At the molecular level, we examined the expression of a total of five adipocyte differentiation markers, namely PPARγ, C/EBPα, C/EBPβ, AP2 and SREBP1. Upon the overexpression of PDZK1IP1, we found that PPARγ, C/EBPα, C/EBPβ and SREBP1 were remarkably downregulated (Figure 2E). However, C/EBPα, C/EBPβ and AP2 were remarkably upregulated upon PDZK1IP1 knockdown (Figure 3E). Interestingly, only C/EBPα and C/EBPβ showed the opposite trend after both overexpression and interference. This suggests that PDZK1IP1 likely influences, directly or indirectly, C/EBPα and C/EBPβ expression to regulate goat subcutaneous preadipocyte differentiation. Moreover, regarding why the expression of PPARγ and SREBP1 was not upregulated after PDZK1IP1 knockdown, we believe that this may be because the regulation of cell signaling networks is complex. Thus, our gain and loss of function experiments indicated that PDZK1IP1 is a repressor of goat subcutaneous preadipocyte differentiation at both the morphological and molecular levels.

The effect of PDZK1IP1 on autophagy is rarely reported. One study showed that the overexpression of PDZK1IP1 inhibited bortezomib-induced autophagy of breast cancer [24]. However, the effect of PDZK1IP1 on the autophagy level of goat preadipocytes has not been reported. As is well known, when autophagy is activated, LC3I will be converted into LC3II, which is localized to autophagosome membranes and is an essential step for autophagosome membrane biogenesis. Therefore, LC3II has been recognized as a biomarker of autophagosome formation [27]. Furthermore, p62 (a protein specifically degraded in lysosomes) can also be used as an autophagy marker besides LC3II [28]. In our study, we found that low and strong expression of PDZK1IP1 attenuated and enhanced LC3II transformation in goat subcutaneous preadipocytes, respectively, suggesting that PDZK1IP1 can promote autophagy initiation. However, both reduced autophagosome degradation and enhanced LC3II conversion can lead to LC3II accumulation. We examined the expression levels of p62 in our next study and observed the downregulation of the p62 protein level after the overexpression of PDZK1IP1 (Figure 4A). Nevertheless, the protein level of p62 was upregulated after interference with PDZK1IP1 (Figure 5A). This suggests that the increase in LC3II levels is not mediated by the retardation of autophagosome degradation. In conclusion, our data indicated that PDZK1IP1 promotes autophagy in goat subcutaneous preadipocytes.

Autophagy, a process of lysosome-dependent degradation, can be broadly divided into nonselective and selective forms. Since the discovery of selective autophagy, including lipophagy and mitophagy [26,31], the relationship between autophagy and adipose metabolism has become an ever-increasing topic of interest. It was previously demonstrated that autophagy increased in the adipose tissue of both obese mice and obese humans [14,32]. The specific knockdown of ATG7 in adipose tissue led to mutant mice becoming leaner and the white adipose tissue content was lower as compared with wild-type mice. By knocking out ATG7 in 3T3-L1 preadipocytes, ATG7 knockdown cells showed reduced TG deposition and the downregulation of markers of adipocyte differentiation [12,13]. Similarly, the specific knockdown of ATG5, which is a protein necessary for autophagy, in mouse MEFs blocked the differentiation of preadipocytes [33]. Previous work has shown that the differentiation of 3T3-L1 is inversely correlated with autophagy. They found that treatment with the autophagy inducer rapamycin inhibited the differentiation of 3T3-L1 preadipocytes [34]. Moreover, the study of autophagy in the regulation of bovine preadipocyte differentiation has been reported [35]. In the present study, we investigated the functions of PDZK1IP1 in goat subcutaneous preadipocytes and found that PDZK1IP1 inhibited preadipocyte differentiation but promoted autophagy initiation and increased autophagic flux. Thus, we speculated that PDZK1IP1 may regulate the differentiation of goat subcutaneous preadipocytes by affecting autophagy.

In the next step, autophagy-specific inhibitor NH_4_CL was used to inhibit autophagy in goat subcutaneous preadipocytes, and the relationship between autophagy and differentiation was further studied. NH_4_CL has been considered as an autophagy inhibitor with the ability to increase the lysosomal pH, thus preventing autophagic protein degradation to block autophagic flux [36,37,38,39]. Firstly, we demonstrated that 20 mM NH_4_CL successfully inhibited autophagy. Then, during goat subcutaneous preadipocytes’ differentiation, we treated these cells with 20 mM NH_4_CL for 48 h after PDZK1IP1 was overexpressed. We observed that PDZK1IP1 overexpression reduced the lipid accumulation at the morphological level by Oil Red O staining and Bodipy staining, However, there was no significant difference in lipid accumulation after the overexpression of PDZK1IP1 in the 20 mM NH_4_CL treatment group (Figure 7B–D). Consistent with the phenotype, the mRNA levels of adipocyte differentiation markers, including PPARγ, C/EBPα and C/EBPβ, were remarkably downregulated after PDZK1IP1 was overexpressed, which could be rescued to normal levels by NH_4_CL treatment (Figure 7E). Similarly, we also observed that only the pattern of C/EBPβ expression was consistent with our speculation. Although the mRNA expression of PPARγ and C/EBPα could be rescued to normal levels by NH_4_CL treatment, the overall levels were higher than in the control group. The regulation of gene expression is often multifactorial, because gene expression regulation is a complex and dynamic process, which involves multi-level regulation, including transcriptional, post-transcriptional, translational and post-translational events. Alternatively, gene expression also may be regulated by one or more signaling pathways. The above data indicate that the inhibition of autophagy can reverse the PDZK1IP1-induced differentiation restraint in goat subcutaneous preadipocytes.

## 5. Conclusions

In summary, our study found that PDZK1IP1 inhibited preadipocyte differentiation, promoted autophagy initiation and increased autophagic flux in goat subcutaneous preadipocytes. In addition, PDZKIP1 inhibited goat subcutaneous preadipocyte differentiation by promoting autophagy. These results provide an essential reference for the further study of PDZK1IP1 in the regulation of goat lipid accumulation. Our study also enriches the knowledge of the regulatory networks of goat preadipocyte differentiation.

## Figures and Tables

**Figure 1 animals-13-01046-f001:**
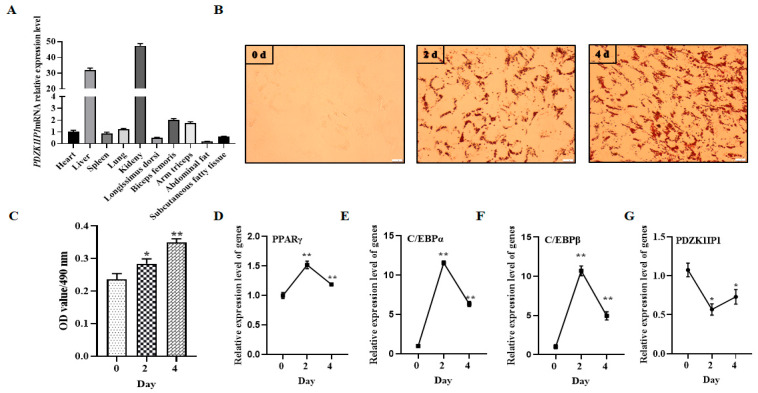
The expression pattern of PDZK1IP1 in goat subcutaneous preadipocytes. (**A**) The expression patterns of PDZK1IP1 in various goat tissue samples. (**B**) Representative images (400×, scale bar = 20 μm) of Oil Red O staining for day 0, 2 and 4. (**C**) Oil Red O staining was performed with 100% isopropanol and absorbance at 490 nm was measured. (**D**–**G**) The expression levels of PPARγ, C/EBPα, C/EBPβ and PDZK1IP1 during the differentiation of goat subcutaneous preadipocytes. The experiments above were all performed at least in triplicate. * indicates *p* < 0.05 in comparison to day 0 group, ** indicates *p* < 0.01 in comparison to day 0 group.

**Figure 2 animals-13-01046-f002:**
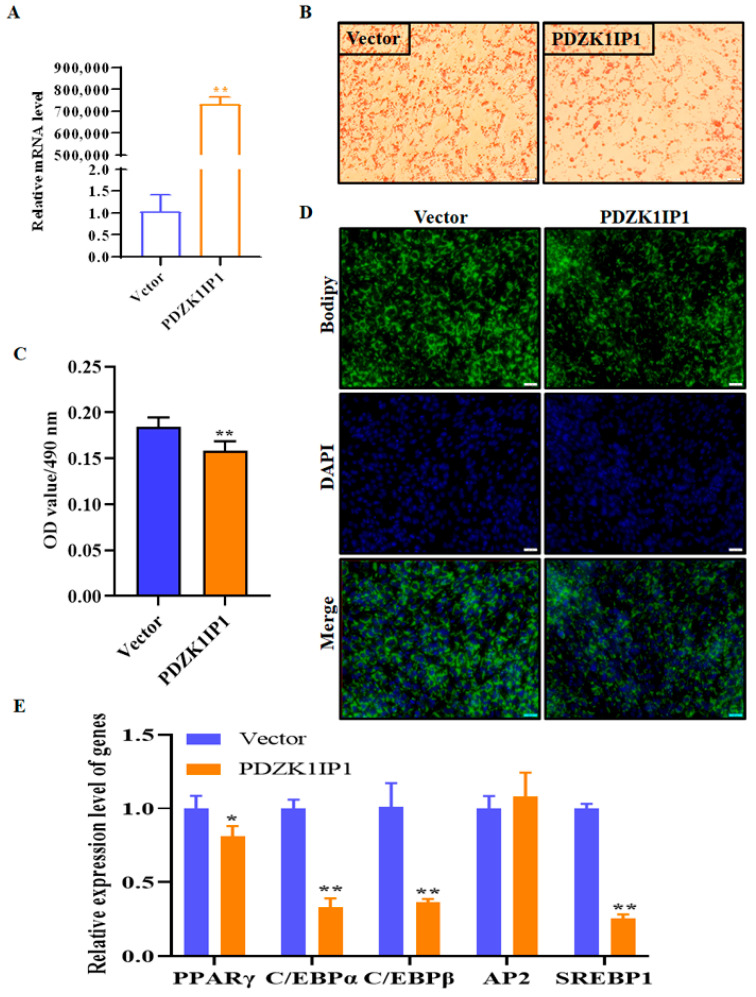
PDZK1IP1 overexpression inhibits the adipogenic differentiation of goat subcutaneous preadipocytes. (**A**) Efficiency of PDZK1IP1 overexpression was assessed by qRT-PCR at 48 h after transfection. (**B**) Representative images of Oil Red O staining of goat subcutaneous preadipocytes and transfection with vector and PDZK1IP1. (**C**) Oil Red O staining was performed with 100% isopropanol and absorbance at 490 nm was measured. (**D**) Representative images of lipid staining by Bodipy; nucleus by DAPI; the overlapping of two channels is shown in merged image. (**E**) The adipocyte differentiation marker gene’s mRNA levels were measured by qRT-PCR analysis. All photos were magnified at 400×, scale bar = 20 μm. The experiments above were all performed at least in triplicate. * indicates *p* values < 0.05, ** indicates *p* values < 0.01.

**Figure 3 animals-13-01046-f003:**
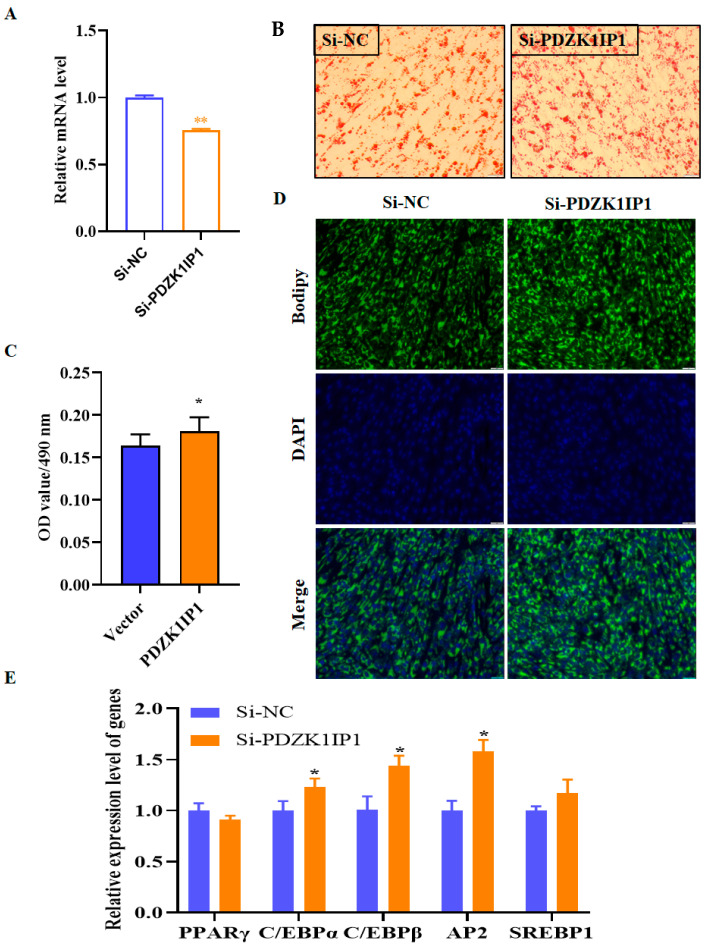
PDZK1IP1 knockdown promotes the adipogenic differentiation of goat subcutaneous preadipocytes. (**A**) Efficiency of PDZK1IP1 knockdown was assessed by qRT-PCR at 48 h after transfection. (**B**) Representative images of Oil Red O staining of goat subcutaneous preadipocytes and transfection with Si-NC and Si-PDZK1IP1. (**C**) Oil Red O staining was extracted with 100% isopropanol and absorbance at 490 nm was measured. (**D**) Representative images of lipid staining by Bodipy; nucleus by DAPI; the overlapping of two channels is shown in merged image. (**E**) The adipocyte differentiation marker gene’s mRNA levels were measured by qRT-PCR analysis. All photos were magnified at 400×, scale bar = 20 μm. The experiments above were all performed at least in triplicate. * indicates *p* values < 0.05, ** indicates *p* values < 0.01.

**Figure 4 animals-13-01046-f004:**
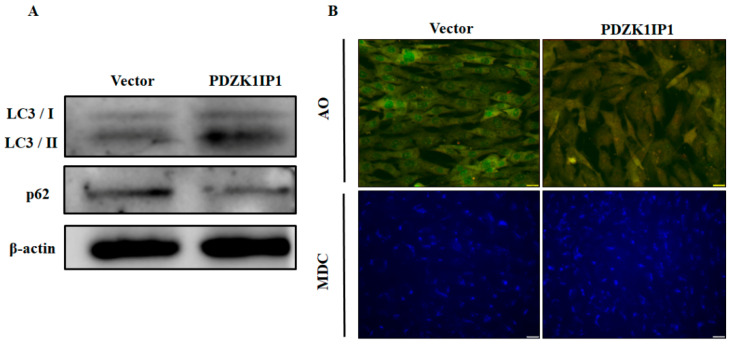
PDZK1IP1 overexpression promotes the autophagy of goat subcutaneous preadipocytes. (**A**) Expression of protein levels of LC3 II/I and p62 detected by Western blot at 48 h after vector or PDZK1IP1 plasmid transfection. β-actin was used as a loading control. (**B**) Images of acridine orange (AO) staining and monodansylcadaverine (MDC) staining in goat subcutaneous preadipocytes as detected using fluorescence microscopy are shown, at 48 h after vector or PDZK1IP1 plasmid transfection. All photos were magnified at 400×, scale bar = 20 μm.

**Figure 5 animals-13-01046-f005:**
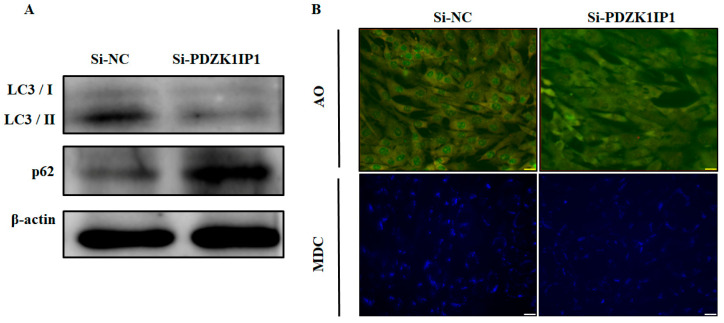
PDZK1IP1 knockdown inhibits the autophagy of goat subcutaneous preadipocytes. (**A**) Expression of protein levels of LC3 II/I and p62 detected by Western blot at 48 h after Si-NC or Si-PDZK1IP1 transfection. β-actin was used as a loading control. (**B**) Images of acridine orange (AO) staining and monodansylcadaverine (MDC) staining in goat subcutaneous preadipocytes as detected using fluorescence microscopy are shown, at 48 h after Si-NC or Si-PDZK1IP1 transfection. All photos were magnified at 400×, scale bar = 20 μm.

**Figure 6 animals-13-01046-f006:**
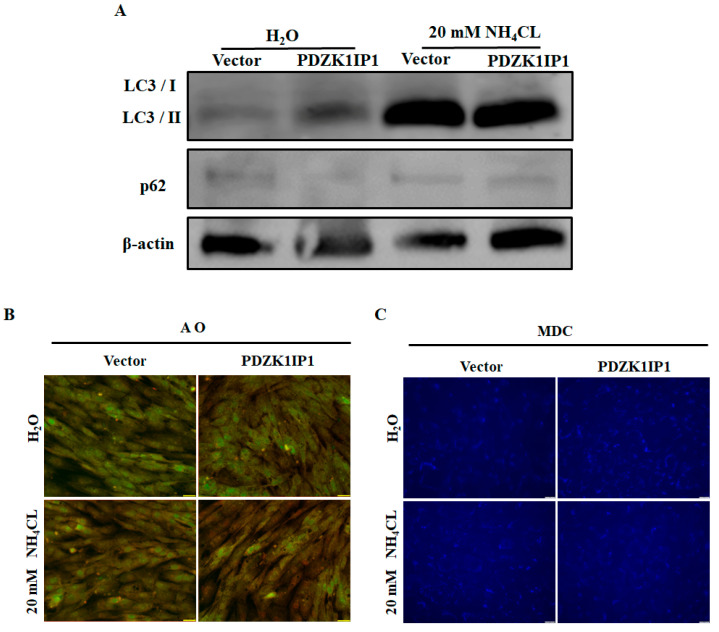
Effect of autophagy inhibitor NH_4_CL on autophagy flux. (**A**) LC3 II/I and p62 protein expression levels in goat subcutaneous preadipocytes treated with 20 mM NH_4_CL for 48 h were measured using Western blot analysis. (**B**–**C**) Images of acridine orange (AO) staining and monodansylcadaverine (MDC) staining in goat subcutaneous preadipocytes as detected using fluorescence microscopy are shown, after goat subcutaneous preadipocytes were treated with 20 mM NH_4_CL for 48 h. All photos were magnified at 400×, scale bar = 20 μm.

**Figure 7 animals-13-01046-f007:**
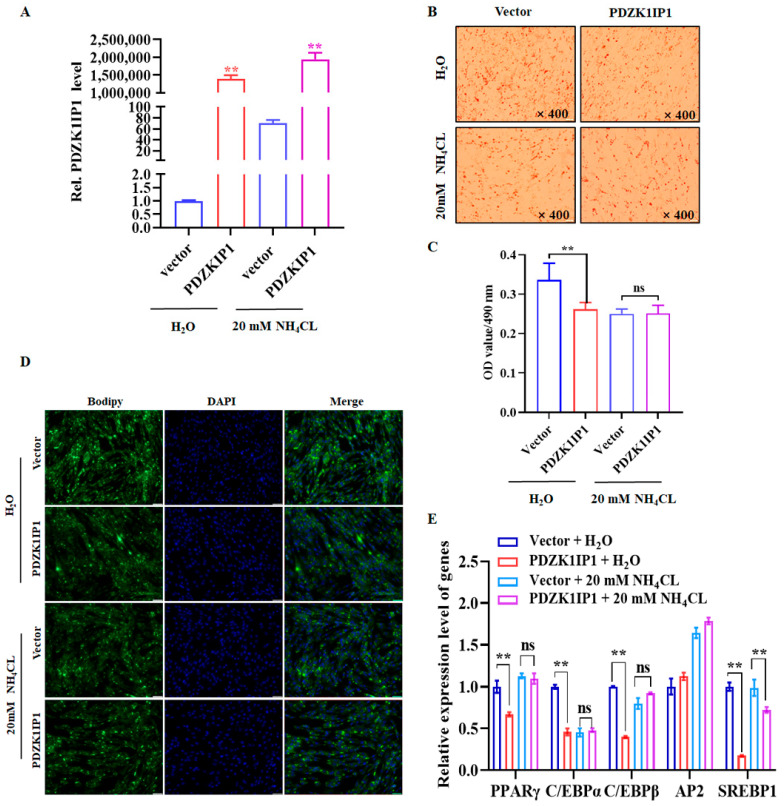
Inhibition of autophagy can rescue PDZK1IP1-induced differentiation restraint in goat subcutaneous preadipocytes. (**A**) Efficiency of PDZK1IP1 overexpression was assessed by qRT-PCR at 48 h after goat subcutaneous preadipocytes were treated with 20 mM NH_4_CL. (**B**) Representative images of Oil Red O staining of goat subcutaneous preadipocytes and transfection with vector and PDZK1IP1. (**C**) Oil Red O staining was performed with 100% isopropanol and absorbance at 490 nm was measured, after goat subcutaneous preadipocytes were treated with 20 mM NH_4_CL for 48 h. (**D**) After goat subcutaneous preadipocytes were treated with 20 mM NH_4_CL for 48 h, representative images of lipid staining by Bodipy were obtained; nucleus by DAPI; the overlapping of two channels is shown in merged image. (**E**) The adipocyte differentiation marker gene’s mRNA levels were measured by qRT-PCR analysis, after goat subcutaneous preadipocytes were treated with 20 mM NH_4_CL for 48 h. All photos were magnified at 400 ×, scale bar = 20 μm. ** indicates *p* values < 0.01, ns indicates no statistical significance.

**Table 1 animals-13-01046-t001:** Primer information for quantitative real-time PCR (qRT-PCR).

Gene Name	Forward Sequence (5′–3′)	Reverse Sequence (5′–3′)
PDZK1IP1	TGTGTTCCTGGTCCTTGTCG	CCTCCTCCTCTGGGATGTTC
C/EBPα	CTCCGGATCTCAAGACTGCC	CCCCTCATCTTAGACGCACC
C/EBPβ	CCGCCTTTAAATCCATGGAA	CTCGTGCTCTCCGATGCTAC
PPARγ	AAGCGTCAGGGTTCCACTATG	GAACCTGATGGCGTTATGAGAC
AP2	TGAAGTCACTCCAGATGACAGG	TGACACATTCCAGCACCAGC
SREBP1	AACATCTGTTGGAGCGAGCA	TCCAGCCATATCCGAACAGC
UXT	GCAAGTGGATTTGGGCTGTAAC	ATGGAGTCCTTGGTGAGGTTGT

## Data Availability

Data are contained within the article.

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
