# Peer review of "PDZK1-Interacting Protein 1(PDZKIP1) Inhibits Goat Subcutaneous Preadipocyte Differentiation through Promoting Autophagy"

_animals, 2023, doi:10.3390/ani13061046_

Round 1
Reviewer 1 Report
This manuscript (2241653) is trying to investigate the function of PDZK1-interacting protein 1 (PDZK1IP1) in goat subcutaneous preadipocytes in vitro. Firstly, the authors examined the effect of PDZK1IP1 on differentiation and autophagy in goat subcutaneous preadipocytes. Authors, then, demonstrated PDZKIP1 inhibits goat subcutaneous preadipocytes differentiation through promoting autophagy. I feel the potential interest of this research but found there are some minor issues.
1. Why do you want to study the role of PDZK1IP1 in goat subcutaneous preadipocytes in vitro?
2. You should add the permission number of animal experiment from the animal experiment committee of your institution in method section.
3. For Oil Red O staining and Bodipy staining, how many cells were seeded on 24-well? Please mention this in the methodology or results section.
4. How did the authors conclude the concentration of oleic acid (50 μM) to be used for the experiment?
5. The clarity of figure, such as Figure 3, is too low. Please correct it.
Author Response
oint-By-Point Responses
Reviewer 1#
- Why do you want to study the role of PDZK1IP1 in goat subcutaneous preadipocytes in vitro?
Response: Thank you very much for your valuable comment. In livestock production, the intramuscular fat (IMF) content is positively related to various aspects of meat quality, including tenderness, and flavor. Similarly, subcutaneous fat is also one of the important factors that influence the taste of meat. Improving meat quality, understanding the mechanisms of regulation of adipogenesis in farm animals is essential. It is known that adipocyte proliferation and differentiation are the basis for the accumulation of lipids. It has been demonstrated that PDZK1IP1 is involved in the occurrence and development of human cancers, and its overexpressed expression is associated with the proliferation of tumor cells. In our previous study, we successfully cloned goat PDZK1P1 gene sequence and demonstrated that PDZK1IP1 overexpression could promote goat subcutaneous preadipocytes proliferation. However, the regulating effects of PDZK1IP1 on regulating lipid deposition in preadipocytes has not been reported. Thus, in this study, we want to explore the role of PDZK1IP1 in regulating subcutaneous preadipocyte lipid deposition in goat. These results could provide the essential data for improving meat quality and understanding the mechanisms of regulation of adipogenesis in farm animals.
- You should add the permission number of animal experiment from the animal experiment committee of your institution in method section.
Response: Thank you very much for your valuable suggestion. All experimental procedures were approved by the Institutional Animal Care and Use Committee of Southwest Minzu University (Chengdu, Sichuan, China) with approval No. 2020086,2020. This section has been added to the method and the text has been revised accordingly (line 97 to line 99).
- For Oil Red O staining and Bodipy staining, how many cells were seeded on 24-well? Please mention this in the methodology or results section.
Response: Thank you very much for your valuable comment and suggestion. For Oil Red O staining and Bodipy staining, cells were seeded in 24-well plates at a density of 1×104 cells per well. This section has been added to the method and the text has been revised accordingly (line 138 to line 139).
- How did the authors conclude the concentration of oleic acid (50 μM) to be used for the experiment?
Response: Thank you for your valuable comment. When the precursor adipocyte culture system of goats was established in the laboratory, the concentration of oleic acid induction solution was screened and 50 μm/L was finally determined as the optimal concentration, which was adopted in the related articles published in the laboratory recently (Li et al., 2022; Xu et al., 2021; xiong et al., 2018).
[1] Li X, Zhang H, Wang Y, Li Y, He C, Zhu J, Xiong Y, Lin Y. RNA-seq analysis reveals the positive role of KLF5 in the differentiation of subcutaneous adipocyte in goats. Gene. 2022; 808:145969.
[2] Xu Q, Wang Y, Li X, Du Y, Li Y, Zhu J, Lin Y. miR-10a-5p Inhibits the Differentiation of Goat Intramuscular Preadipocytes by Targeting KLF8 in Goats. Front Mol Biosci. 2021; 8:700078.
[3] Xiong Y, Xu Q, Lin S, Wang Y,Lin YQ, Zhu JJ. Knockdown of LXRα inhibits goat intramuscular preadipocyte differentiation. Int J Mol Sci, 2018, 19(10): 3037.
- The clarity of figure, such as Figure 3, is too low. Please correct it.
Response: Thank you very much for your valuable comment and suggestion. Figure 3 has been adjusted for better clarity and had been revised accordingly.

Reviewer 2 Report
This works shows that PDZK1IP1 inhibited goat subcutaneous preadipocytes differentiation through promoting autophagy, when overexpressed or silenced PDZK1IP1, the mRNA and protein levels of proliferation or differentiation related genes were changed, which were further validated by western blot and qRT-PCR. subsequently, Oil Red O staining, semi-quantitative assessment of Oil Red O content and Bodipy staining also confirmed that PDZK1IP1 inhibited preadipocyte differerniation. the experimental design and results are complete, however I have few suggestions:
-The P value would be better written in italics;
-The WB result of Fig.6, LC3I couldnt be seen clearly,maybe it can be presented better;
-Almost all bars dont show the number of biological repeats, you must indicate n=?
Author Response
Reviewer 2#
- The P value would be better written in italics
Response: Thank you very much for your suggestion. We have corrected the written of “P” in the text (line 183 to line 184; line 216 to line 217; line 239 to line 240 and line 263 to line 264).
- The WB result of Fig.6, LC3I couldn’t be seen clearly, maybe it can be presented better
Response: Thank you very much for your suggestion. Fig. 6 has been adjusted for better clarity to make LC3 I clear and the text had been revised accordingly.
- Almost all bars don’t show the number of biological repeats, you must indicate n=?
Response: Thank you very much for your valuable comment and suggestion. The number of biological repeats in all bars is three. This section has been added to the method and the text has been revised accordingly (line 184 to line 185).

Reviewer 3 Report
Comments to Author:The manuscript submitted provides information on the PDZK1-interacting protein 1(PDZKIP1) inhibits goat 2 subcutaneous preadipocytes differentiation through promoting 3 autophagy.
The methods are described in suitable detail albeit animal related information is missing (body weight, diet, housing, conditions that show that the animals were healthy). Some results were repeated in the discussion section using identical phrases.
Specific comments.
Title: suggest revise to ‘PDZK1-interacting protein 1(PDZKIP1) inhibits goat 2 subcutaneous preadipocytes differentiation through promoting 3 autophagy’.
L48: suggest revising to ‘because they are an important factor that impacts meat qualities’.
L49: suggest combining these 2 sentences ‘because they are an important factor that impacts meat qualities, such as, high intramuscular fat content…’
L50: suggest revising to ‘However, increased subcutaneous fat..’
L51: suggest revise ‘However, increased subcutaneous fat content can reduce the lean meat ratio of the carcass, reducing the economic value of the meat’.
L53: suggest revise ‘…agricultural economy’.
L56: need to include a space. Macromolecules [11]
L56: suggest revise to ‘Has been shown..”
L67: suggest remove the word etc.
L73: could you please confirm you are talking about more than one study? ‘In our previous studies…’ If talking about one study, I suggest revise to ‘In our previous study…’
L83: suggest revise to ‘Thus, the current study was to investigate the underlying…’
L84-85: suggest revise to ‘Our study focused on the effects of PDZK1IP1 on…’
L86-88: suggest remove ‘The results of this study will help to better understanding the biological functions of PDZK1IP1 in goat subcutaneous preadipocytes and improve its molecular mechanism studies’ from lines 86 to 88 or 18 to 20, because they are identical.
L92: suggest adding, average body weight prior to slaughter, animals had ad libitum access to the same diet, water?
L92: suggest revise to ‘All experimental procedures were approved by the University of Southwest Minzu (Chengdu, Sichuan, China) Institutional Animal Care and Use Committee.
L94: suggest revise to ‘A total of three Jianzhou Daer goats (Capra hicus) male, one year old, were slaughtered. Tissue samples included heart, liver, spleen, lungs, kidney, longissimus dorsi, biceps femoris, arm triceps, abdominal fat and subcutaneous fatty tissue were harvested and immediately frozen in liquid nitrogen.
L94: suggest including where the goats were slaughtered.
L102: suggest revise to ‘Briefly, in our previously study,…”
L176: suggest adding space. …’method [26].
L182: suggest revise to '… In every figure…’
L183: suggest changing format for italic. (* p < 0.05; ** p < 0.01).
L189: suggest revise to ‘The expression level of PDZK1IP1 was higher in kidney and lower in abdominal fat compared to expression level in heart’.
L190: suggest rephrasing. Here, low expression of PDZK1IP1 was also observed in subcutaneous adipose tissue which is our concern (Figure 1A).
L360: suggest revise to ‘However, we still do not know how the effect of PDZK1IP1 on goat subcutaneous preadipocytes differentiation is.
L366: I suggest you review the results and discussion sections to avoid repeat the same phrases. Another example is …’but lower in subcutaneous adipose tissue which is our concern (Line 366) and low expression of PDZK1IP1 was also observed in subcutaneous adipose tissue which is our concern (Line 191).
L399: suggest revise to ‘We examined the expression levels of p62 in the next study and observed…’.
L401: I suggest you rewriting this sentence., it will make it easier to read. ‘Of course, observed an upregulation of p62 protein level when PDZK1IP1 was knocked down’.

Author Response
eviewer 3#
- Title: suggest revise to ‘PDZK1-interacting protein 1(PDZKIP1) inhibits goat 2 subcutaneous preadipocytes differentiation through promoting 3 autophagy’.
Response: Thank you very much for your valuable suggestion. The title has been revised accordingly (line 2 to line 4).
- L48: suggest revising to ‘because they are an important factor that impacts meat qualities’.
Response: Thank you very much for your valuable suggestion. The sentence has been revised accordingly (line 47 to line 50).
- L49: suggest combining these 2 sentences ‘because they are an important factor that impacts meat qualities, such as, high intramuscular fat content…’
Response: Thank you very much for your valuable comment and suggestion. These 2 sentences have been combined accordingly and the text has been revised accordingly (line 47 to line 50).
- L50: suggest revising to ‘However, increased subcutaneous fat..’
Response: Thank you very much for your valuable suggestion. The sentence has been revised accordingly (line 50 to line 52).
- L51: suggest revise ‘However, increased subcutaneous fat content can reduce the lean meat ratio of the carcass, reducing the economic value of the meat’.
Response: Thank you very much for your valuable suggestion. The sentence has been revised accordingly (line 50 to line 52).
- L53: suggest revise ‘…agricultural economy’.
Response: Thank you very much for your valuable suggestion. The sentence has been revised accordingly (line 52 to line 53).
- L56: need to include a space. Macromolecules [11]
Response: Thank you very much for your valuable suggestion. The sentence has been revised accordingly (line 54 to line 56).
- L56: suggest revise to ‘Has been shown..”
Response: Thank you very much for your valuable suggestion. The sentence was modified to ‘Has been shown..’. And the text has been revised accordingly (line 56 to line 57).
- L67: suggest remove the word etc.
Response: Thank you very much for your valuable suggestion. The word etc. has been removed and the text has been revised accordingly (line 65 to line 67).
- L73: could you please confirm you are talking about more than one study? ‘In our previous studies…’ If talking about one study, I suggest revise to ‘In our previous study…’
Response: Thank you very much for your valuable comment and suggestion. The text has been revised accordingly (line 73 to line 75).
- L83: suggest revise to ‘Thus, the current study was to investigate the underlying…’
Response: Thank you very much for your valuable suggestion. The sentence has been revised accordingly (line 82 to line 83).
- L84-85: suggest revise to ‘Our study focused on the effects of PDZK1IP1 on…’
Response: Thank you very much for your valuable suggestion. The sentence has been revised accordingly (line 83 to line 85).
- L86-88: suggest remove ‘The results of this study will help to better understanding the biological functions of PDZK1IP1 in goat subcutaneous preadipocytes and improve its molecular mechanism studies’ from lines 86 to 88 or 18 to 20, because they are identical.
Response: Thank you very much for your valuable suggestion. The sentence from lines 86 to 88 has been removed accordingly.
- L92: suggest adding, average body weight prior to slaughter, animals had ad libitum access to the same diet, water?
Response: Thank you very much for your valuable suggestion. The body weight of goats was approximately 50 kg. In addition, these goats were given ad libitum access to the same diet and water before slaughter. This section has been added to the method and the text has been revised accordingly (line 90 to line 92).
- L92: suggest revise to ‘All experimental procedures were approved by the University of Southwest Minzu (Chengdu, Sichuan, China) Institutional Animal Care and Use Committee.
Response: Thank you very much for your valuable suggestion. The sentence has been revised accordingly (line 97 to line 99).
- L94: suggest revise to ‘A total of three Jianzhou Daer goats (Capra hicus) male, one year old, were slaughtered. Tissue samples included heart, liver, spleen, lungs, kidney, longissimus dorsi, biceps femoris, arm triceps, abdominal fat and subcutaneous fatty tissue were harvested and immediately frozen in liquid nitrogen.
Response: Thank you very much for your valuable suggestion. The sentence has been revised accordingly (line 92 to line 95).
- L94: suggest including where the goats were slaughtered.
Response: Thank you very much for your valuable suggestion. The slaughter procedure was conducted at College of Animal Science and Veterinary, Southwest Minzu University. This section has been added to the method and the text has been revised accordingly (line 96 to line 97).
- L102: suggest revise to ‘Briefly, in our previously study,…”
Response: Thank you very much for your valuable suggestion. The sentence has been revised accordingly (line 102 to line 104).
- L176: suggest adding space. …’method [26].
Response: Thank you very much for your valuable suggestion. The space has been added accordingly (line 176 to line 177).
- L182: suggest revise to '… In every figure…’
Response: Thank you very much for your valuable suggestion. The sentence has been revised accordingly (line 183 to line 184).
- L183: suggest changing format for italic. (* p < 0.05; ** p < 0.01).
Response: Thank you very much for your suggestion. We have corrected the written of “P” in the text (line 183 to line 184; line 216 to line 217; line 239 to line 240 and line 263 to line 264).
- L189: suggest revise to ‘The expression level of PDZK1IP1 was higher in kidney and lower in abdominal fat compared to expression level in heart’.
Response: Thank you very much for your valuable comment and suggestion. The sentence has been revised accordingly (line 191 to line 193).
- L190: suggest rephrasing. Here, low expression of PDZK1IP1 was also observed in subcutaneous adipose tissue which is our concern (Figure 1A).
Response: Thank you very much for your suggestion. The sentence has been rewritten into ‘In addition, PDZK1IP1 was also expressed in subcutaneous adipose tissue (Figure 1A)’. The text has been revised accordingly (line 193 to line 194).
- L360: suggest revise to ‘However, we still do not know how the effect of PDZK1IP1 on goat subcutaneous preadipocytes differentiation is.
Response: Thank you very much for your valuable suggestion. The sentence has been revised accordingly (line 362 to line 363).
- L366: I suggest you review the results and discussion sections to avoid repeat the same phrases. Another example is …’but lower in subcutaneous adipose tissue which is our concern (Line 366) and low expression of PDZK1IP1 was also observed in subcutaneous adipose tissue which is our concern (Line 191).
Response: Thank you very much for your valuable suggestion. We have checked the manuscript carefully and rewritten repeated sentences in the results and discussion sections. For example, the first sentence was modified to ‘Thus, we examined the expression of PDZK1IP1 in multiple goat tissues and found it to be expressed in all tested tissues, including subcutaneous adipose tissue’ (line 365 to line 367). The another sentence was modified to ‘Furthermore, p62 (a protein specifically degraded in lysosomes) can also be used as an autophagy marker other than LC3II’ (line 395 to line 396).
- L399: suggest revise to ‘We examined the expression levels of p62 in the next study and observed…’.
Response: Thank you very much for your valuable suggestion. The sentence has been revised as suggested. (line 401 to line 402).
- L401: I suggest you rewriting this sentence., it will make it easier to read. ‘Of course, observed an upregulation of p62 protein level when PDZK1IP1 was knocked down’.
Response: Thank you very much for your valuable comment and suggestion. The sentence has been rewritten into ‘Nevertheless, the protein level of p62 was upregulated after interference with PDZK1IP1’. The text has been revised accordingly (line 403 to line 404).
Round 2
Reviewer 3 Report
L49: suggest revise to “such as, high intramuscular fat content…”
Corrections have been made with no other suggested revisions found.
Author Response
Reviewer 3#
- L49: suggest revise to “such as, high intramuscular fat content…” Corrections have been made with no other suggested revisions found.
Response: Thank you very much for your valuable suggestion. The sentence has been revised accordingly (line 49 to line 50).
